# Host Cell Signatures of the Envelopment Site within Beta-Herpes Virions

**DOI:** 10.3390/ijms23179994

**Published:** 2022-09-01

**Authors:** Hana Mahmutefendić Lučin, Gordana Blagojević Zagorac, Marina Marcelić, Pero Lučin

**Affiliations:** Department of Physiology, Immunology and Pathophysiology, Faculty of Medicine, University of Rijeka, 51000 Rijeka, Croatia

**Keywords:** beta-herpesviruses, cytomegalovirus, cytomegalovirus envelopment, assembly compartment, beta-herpesvirus virions, proteome

## Abstract

Beta-herpesvirus infection completely reorganizes the membrane system of the cell. This system is maintained by the spatiotemporal arrangement of more than 3000 cellular proteins that continuously adapt the configuration of membrane organelles according to cellular needs. Beta-herpesvirus infection establishes a new configuration known as the assembly compartment (AC). The AC membranes are loaded with virus-encoded proteins during the long replication cycle and used for the final envelopment of the newly formed capsids to form infectious virions. The identity of the envelopment membranes is still largely unknown. Electron microscopy and immunofluorescence studies suggest that the envelopment occurs as a membrane wrapping around the capsids, similar to the growth of phagophores, in the area of the AC with the membrane identities of early/recycling endosomes and the trans-Golgi network. During wrapping, host cell proteins that define the identity and shape of these membranes are captured along with the capsids and incorporated into the virions as host cell signatures. In this report, we reviewed the existing information on host cell signatures in human cytomegalovirus (HCMV) virions. We analyzed the published proteomes of the HCMV virion preparations that identified a large number of host cell proteins. Virion purification methods are not yet advanced enough to separate all of the components of the rich extracellular material, including the large amounts of non-vesicular extracellular particles (NVEPs). Therefore, we used the proteomic data from large and small extracellular vesicles (*l*EVs and *s*EVs) and NVEPs to filter out the host cell proteins identified in the viral proteomes. Using these filters, we were able to narrow down the analysis of the host cell signatures within the virions and determine that envelopment likely occurs at the membranes derived from the tubular recycling endosomes. Many of these signatures were also found at the autophagosomes, suggesting that the CMV-infected cell forms membrane organelles with phagophore growth properties using early endosomal host cell machinery that coordinates endosomal recycling.

## 1. Introduction

Beta-herpesviruses infect a large proportion of the human population and are associated with a variety of pathophysiological conditions [1]. They are DNA viruses with a large genome that encodes a relatively large number of gene products for the construction of new viral progeny and the establishment of a complex series of interactions with infected cells [1]. The genes are expressed in a sequence that is usually divided into the immediate early (IE), early (E), and late (L) phases of expression. After the replication of viral DNA, viral capsids are assembled in the nucleus and exported to the cytosol through a temporary envelopment with the nuclear membrane (primary envelopment). The exported capsids are embedded with tegument proteins and acquire a final membrane envelope in a process called secondary envelopment (reviewed in [1,2,3]). This process takes place within the large structure formed by membranous organelles in the cytosol of the infected cell called the assembly compartment (AC). Secondary envelopment and egress are critical phases in the beta-herpesvirus replication cycle and are exceptional targets for antiviral drug development. Therefore, understanding the biogenesis and identity of the membranes used for envelopment and the pathways used for egress are essential for understanding beta-herpesvirus pathogenesis and the development of antivirals. 

Most knowledge of beta-herpesvirus assembly comes from studies of human cytomegalovirus (human herpesvirus 5, HHV-5, or HCMV), an important human pathogen with several pathophysiological implications [4], and to a lesser extent murine cytomegalovirus (murid herpesvirus 1, MuHV-1, or MCMV), an important model for studying beta-herpesvirus biology [5]. Nevertheless, the identity of the envelopment organelle remains unknown, and the mechanism of egress of the newly formed virions is unclear [1,2,3]. One reason for this is the extreme complexity of CMV interaction with the host and the insufficient understanding of the physiology of the membrane system. The high-throughput approaches have incredibly increased this complexity, and cytoplasmic processes have been found to involve the interaction of CMV-encoded functions with nearly 3000 host cell gene products [6,7]. Secondary envelopment can occur in membrane organelles that combine the functional domains present at the different organelles within the organelle architecture of the uninfected cell. These adaptations include the redirection, retrieval, and maintenance of viral glycoproteins (envelope proteins) in the appropriate membrane composition; recruitment or retrieval of tegument proteins; acquisition of the ability to grow, wrap, and seal around capsids to form infectious virions; migration toward the PM; and acquisition of the secretory ability or toward the secretory organelle that can carry infectious virions out of the cell. However, it is not known whether these adaptations occur in a single envelopment compartment or sequentially during the envelopment process. Over the past two decades, many studies have attempted to determine the identity of the envelope compartment, but most of our knowledge is still based on electron microscopy (EM) studies [1,2]. The construction of the fluorescent capsids proved to be a difficult task for beta-herpesviruses [8], and no true record of the envelopment process has been demonstrated.

Our hypothesis was that the valuable information about the envelopment site must be hidden in the virions and that the host cell proteins that define the identity and associated functions of the membrane domain used for envelopment should be incorporated into the virions and detected using a sufficiently sensitive technique. Thanks to significant advances in mass spectrometry (MS), it is now possible to accurately identify the host cell proteins in the virions. Several studies [9,10,11,12,13,14,15] have already presented proteomic analyses of conventionally purified virion preparations. However, the large number of host cell proteins identified in these preparations does not allow accurate conclusions about the identity of the envelopment organelle. Clearly, more knowledge of the extracellular content released from infected cells is needed to make use of these data. Recent advances in the field of extracellular vesicles have shown that the extracellular content is more complex [16,17,18], which should be considered if we are to infer the identity of the envelopment organelle by analyzing the host cell proteins that remain as signatures in the virions. In this review, we reanalyze the available proteomic data of the virion preparations from the literature through the filters of the proteomic data on extracellular content. The goal is to identify the cellular proteins in the virions that can determine the identity of the envelopment compartment. In this approach, the cellular proteins identified in various forms of extracellular material are rigorously eliminated, which may be a limitation because functional proteins present in the envelopment organelle may also be eliminated. However, those that are not eliminated remain as reliable host cell signatures that can determine the identity of the envelopment organelle.

## 2. Beta-Herpesvirus Secondary Envelopment

Secondary envelopment occurs within the AC, a typical large cytoplasmic membranous organelle cluster established in beta-herpesvirus-infected cells (Figure 1A), as demonstrated in HCMV [19,20,21,22,23] and MCMV [24,25,26,27] studies and presented in several reviews [1,2,3]. The inner area (inner AC) contains many vesicular and tubular elements derived from early endosomes (EEs), the endosomal recycling compartment (ERC), and the trans-Golgi network (TGN). These organelles are surrounded by several layers of membranous elements (outer AC), which contain the modified Golgi stacks and trans-Golgi elements that project toward the inner area. The inner/outer AC configuration is established early in infection [21,26,27,28,29,30,31,32,33] and is initiated by the extensive displacement of the Golgi from the cell center and its realignment in such a way that the trans side accumulates and expands around the cell center along with the EE-ERC-derived elements [21,24,26]. The endoplasmic reticulum (ER), late endosomes (LEs), and lysosomes appear to be relocated to the cell periphery [1,2,19,20,26], but it is unclear whether ER- and LE-derived elements contribute to the inner AC [19,20]. Vacuolized multivesicular membrane elements, likely representing multivesicular endosomes (also known as multivesicular bodies, MVBs) or fused MVBs and autophagosomes (also known as amphisomes), have been frequently observed at the periphery of the inner AC adjacent to the outer AC membrane elements [34]. Vacuolized membrane structures have also been observed within the inner AC [34].

Our current understanding of cytoplasmic envelopment is based mainly on EM studies, and the sequence is shown schematically in Figure 1B. CMV generates three types of capsids (first A-, then B-, and finally C-type capsids) during nuclear maturation [35,36]. Fully packed C-type capsids containing viral DNA are released from the nucleus and adopt the final envelope at the membrane organelles inside the AC. These organelles concentrate viral glycoproteins (called envelope proteins) and condense viral-encoded proteins at the cytoplasmic surface, forming the tegument of the virion. The migrating C-type capsids embed into the tegument and undergo the process of envelopment, which is uniformly distributed throughout the AC, and no preferred site for envelopment has been identified [34]. Scanning transmission electron microscopy (STEM) showed that the HCMV envelopment occurs at the tubular structures that provide sufficient membranes to wrap around the tegumented capsids in the form of a double membrane wrapping (Figure 1B(a)) [2,34,37,38]. However, EM studies also showed the enveloped capsids within the large multivesicular compartments, particularly at later stages of infection, and even suggested that tegumented capsids bud into these large multivesicular compartments (Figure 1B(b)) [34,35,36,37]. Enveloped capsids within a single membrane are released from infected cells as infectious virions [34,37,38]. In addition to capsids containing viral DNA (C-type), the infected nucleus also releases many capsids without DNA (B-type capsids), which can also be enveloped and released from the cell as Non-Infectious Enveloped Particles (NIEPs) [36]. The early EM studies on HCMV envelopment showed that the progeny of virions acquire the envelope at the tubular endosomes [39] and that tubular elements dominate in the images of the AC [40]. A study using fluorescent HCMV particles showed that only a small fraction of them undergo centrifugal long-distance movements, suggesting that egress is not very effective and that the release of the virus is a rather rare event [8].

## 3. Host Cell Signatures within Virions

CMV envelopment, either the membrane wrapping around the virions or the virions budding into membranous organelles, must involve the incorporation of the host cell proteins into the virions. Thus, by analyzing virion composition, we can learn about the origin of the membrane domain used for envelopment and the biochemical requirements for the envelopment process (Figure 1C). However, because the virions are released from the cell together with extracellular vesicles (EVs), the analysis of the host cell signatures in the virions faces all the challenges that are strongly debated in the field of EVs. As discussed in a recent review [41], every cell generates various EVs using diverse mechanisms. Most EVs are developed directly at the PM or in the intracellular compartments, mainly MVBs and LEs, and are released from the cell through their fusion with the PM (Figure 2A). The heterogeneity of EVs is still not well understood and there is no consensus on their biogenesis and molecular composition [41]. EVs can also be released from the cell after autophagosomes have fused with MVB/LEs to form structures known as amphisomes, which deliver autophagocytosed cellular components to lysosomes for degradation but can also fuse with the PM and release the contents outside the cell (Figure 2A). This process, termed secretory autophagy, can contribute cellular contents to the secreted material in addition to increasing the heterogeneity of the extracellular material released. Thus, a major challenge in this field is the optimization of methods to isolate, separate, and characterize the different subpopulations of EVs.

Several recent studies [16,17,18] have taken steps toward increasing the resolution of EVs’ purification and demonstrated that a large amount of secreted material is released from the cell by amphisomes as non-enveloped extracellular particles (NVEPs) that highly overlap in the content with large (l) EVs (*l*EVs) and small (s) EVs (*s*EVs) (Figure 2A,B). NVEPs [16] or exomeres [17,18] represent extracellular, non-membranous molecular assemblies released from cells after cytoplasmic components are engulfed in autophagosomes and fused with MVBs to form amphisomes. Most of the released proteins are the components of large cytoplasmic protein complexes, but components of the membranous organelles (ER, endosomes, and Golgi) can also be found.

The progress in the field of EVs has been paralleled by the efforts to analyze the extracellular content of the herpesvirus-infected cells, including the HCMV [9,10,11,12,13,14,42] and HSV-1 [15] virions, NEIPs of HSV-1 [15], and exosomes released from HCMV- [10] and HSV-1- [15] infected cells. We used the publicly available proteomes of these studies with the aim of identifying the host cell signatures that can reveal the organelle used for envelopment. As expected, these proteomes contained a huge number of host cell proteins, which made it challenging to access the host cell proteins that are indeed packed within the virion envelope. A significant effort was made in a recent study by the Mathias lab [10], which first separated the host cell exosomes and microvesicles from the extracellular virions and then established a protocol to analyze the host proteome in the isolated virions after brief proteinase K treatment. This treatment significantly reduced the number of identified proteins nevertheless, some host cell markers remained that are difficult to explain, such as calnexin or calreticulin, because they are not expected to be membrane-associated or cytosolic markers. Therefore, we matched the available data about the composition of the *s*EVs, *l*EVs, and NVEPs provided in the studies from the Coffey lab [16,17] with the content of the virion preparations. We used the proposed extracellular vesicle association of the cellular markers provided in the Jeppesen et al. study (Figure 2C) [16] to explore the content identified within the HCMV and HSV-2 virion preparations.

As demonstrated in Figure 2C, all the virion preparations were enriched not only in cellular proteins that are associated with classical exosomes, but also in proteins weakly associated with classical exosomes, absent from classical exosomes, or even absent from any type of *s*EVs. The canonical markers of classical exosomes (CD63, CD81, CD9) were mostly cleared by the proteinase K treatment [10]), whereas none of the markers of CD63-only positive exosomes [16] were found (Figure 2C), indicating a minor contribution of exosomes and suggesting a different location for the envelopment from the site of exosome biogenesis. In contrast, some markers that are absent in classical exosomes, notably metabolic enzymes; nucleosome components; annexins 1, 2, and 5; some RNA-binding proteins; components of the cytoskeleton; 14-3-3; VDAC; and calreticulin, were highly enriched in the shaved virions (Figure 2C). Altogether, this analysis demonstrates that the content of the virion preparation highly overlaps with *s*EVs, which mainly include classical exosomes, and *l*EVs, which mainly include microvesicles and non-classical exosomes [16]. The presence of proteins that are never found in *s*EVs also indicates that the content of the virions highly overlaps with the content of NVEPs, which also highly overlap with the content of *s*EVs and *l*EVs. Therefore, the virion preparations used for proteomic analysis, which were prepared by density-gradient separation techniques, contained a substantial amount of EV and NVEP material (Figure 2B). 

### 3.1. Analysis Strategy of the Literature Data

To access the host cell signatures within the virions that may be specific to the site of envelopment, it was essential to subtract the cellar proteins that were present in significant amounts in *l*EVs, *s*EVs, and NVEPs (Figure 2D). To this end, we used the detailed proteomic data from the study by Jeppesen et al. [16] and analyzed the proteomic data of the virions through the sEV, *l*EV, and NVEP filters.

We analyzed the six available proteomes for the presence of the host cell proteins that could be a specific marker for certain stages of membrane system maturation. We analyzed the proteins that are transmembrane proteins with a relatively well-defined itinerary within the normal endomembrane system, either as cargo molecules or as membrane proteins that serve as identity markers for membrane flux. These proteins could not be captured as cytosolic components, which is advantageous for analysis, but they circulate through the membrane compartments, which may be disadvantageous in determining the identity of a compartment because the reconfiguration of the membrane system can be expected in CMV-infected cells. We also analyzed the proteins that are recruited to the cytosolic surface of a membrane compartment, thereby defining the identity of the membrane and representing the zip codes for proper membrane flux, altering the biochemical properties of a membrane, or retaining other proteins (known as cargo). Their location on the cytosolic surface is advantageous because they can be well retained in the virions during preparation procedures (e.g., proteinase K treatment) but they can be trapped in cytosolic complexes (e.g., Rab proteins), which can be a disadvantage during analysis. Some of these proteins are known to shuttle between compartments, and some of them are recruited to membranes at very specific stages of membrane maturation and to distinct membrane domains.

The cellular proteins repeatedly found in different proteomes were considered significant. The proteins identified in proteome 1 [10] from the untreated virion preparations were labeled as “unshaved” and from the proteinase K-treated preparations as “shaved,” as described in the original reference. The proteins consistently detected in the “shaved” virions by two MS methods are considered significant candidates. Different color codes were used to distinguish the proteins found in the unshaved and shaved virion preparations by one or two MS methods. The significance of the proteins detected in proteome 2 [11] was also represented by different colors depending on whether they were detected in one, two, or three biological replicates.

After the host cell proteins were identified in the virion preparations, their presence in the virions was assessed by the NVEP, sEV, and *l*EV filters according to their abundance in the specific proteomes provided by the Jeppesen et al. study [16]. Proteomes from two cell lines were used to evaluate the presence in NVEPs. The abundance was classified as absent if a protein was not detected in the proteomic analysis or if it was below the average MS signal. If the amount detected was 0–2 log_2_ above the average signal, the abundance was classified as very low, as was the case in the Jeppesen et al. study [16]. Amounts 2–3 log_2_ above the average MS signal were classified as low, 3–4 log_2_ as moderate, 4–5 log_2_ as high, and >5 log_2_ as very high. Proteins that were abundant in the virion preparation but also above the threshold of 2 × log_2_ in NVEPs, *s*EVs, and *l*EVs were excluded as potential host cell signatures in the virions because they may be present in NVEPs, *s*EVs, and *l*EVs in the extracellular fraction containing the virions used for proteomic analysis (Figure 2D).

### 3.2. Membrane-Trafficking Cargo Proteins

Cargo proteins are important markers that have been used over the years to learn about endosomal transport. They pass through different compartments within the membrane system and are usually retained in the intracellular localization with low exit rates. Cargo proteins are anchored in the membrane and therefore, when incorporated into the virions, represent the membrane fraction that becomes entrapped in the virions during envelopment, as well as indicate the transport pathway used by the virus for the envelopment process. The cargo proteins identified in the virion preparations were grouped according to the known transport pathways in the endosomal system (Figure 3).

The virion preparations contained proteins that are internalized both clathrin-independently (CIE) and clathrin-dependently (CDE cargo) (Figure 3). Integrins and G protein-coupled receptors (CPCRs) were grouped separately because they can be internalized by both CIE and CDE mechanisms. All of these proteins have different itineraries for endosomal sorting after endocytic uptake into the EEs and are returned to the PM by sorting into different recycling routes or diverted to the LEs and lysosomal degradation. A group of cargo proteins is sorted in the EEs and transported to the Golgi, which is called a retrograde route. The pathways of the known recycling and retrograde cargo proteins are shown in Figure 4. The recycling and retrograde pathways in the EEs are pleiotropic [44,56,57,58] and involve the development of tubular extensions at each stage of EE maturation unless the recycling and retrograde cargo is separated from the degrading cargo (Figure 4). Tubular extensions of the EEs generate transport intermediates that carry cargo directly to the PM (recycling) and Golgi (retrograde pathway) or travel to the cluster of pericentriolar tubular compartments known as the endosomal recycling compartment (ERC) for another round of sorting before returning to the PM or being diverted to the Golgi. This pleomorphic tubular traffic is apparently adapted to the type of cargo and final destination and requires the use of a complex array of sorting mechanisms (i.e., sorting nexins and retrieval complexes), a substantial portion of the Rab protein arsenal in cells, and the involvement of Arf GTPases. The major sites of recruitment of the Rab and Arf proteins, sorting nexins (SNXs), and retrieval complexes are shown in Figure 4.

The virion preparations were enriched in CIE cargo proteins (Figure 3, proteomes 1–6). Most of these proteins, which are enriched in PM fractions [95,96], were detected to varying degrees in NVEPs and were highly enriched in *l*EVs and *s*EVs (Figure 3, proteome 7). After internalization from the PM, the CIE proteins are collected together with the CDE proteins in the Rab5-positive EEs but at this stage, they are sorted into two main pathways [43,44,66]. One subset (CD44, CD98, and CD147) migrates directly from Rab5 endosomes into Rab22-dependent endosomal recycling tubules and does not reach the EEA1-positive compartments. The other subset (MHC-I, CD55, and CD59) returns to the PM indirectly via EEA1-positive endosomes or is sorted into lysosomes for degradation. These proteins are retained in the endosomes of HCMV- [30] and MCMV- [27,97] infected cells. Since MHC-I proteins were not enriched in NVEPs, *l*SVs, and *s*EVs (Figure 3, proteome 7), they can be considered as a cargo protein signature embedded in the virions. Their presence suggests that tubular membranes derived from EEs that recycle CIE cargo are used for envelopment. 

The virion preparations were also rich in integrins (Figure 3, proteomes 1–6), which can use CIE and the CDE route for entry and may use multiple routes for recycling [54]. However, integrins were also highly enriched in *l*EVs and *s*EVs (Figure 3, proteome 7), and therefore cannot be considered a reliable cell signature within the virions.

The virion preparations were enriched in transferrin receptor (TFRC), a well-known CDE cargo; however, TFRC was also enriched in EVs and NVEPs (Figure 3, proteome 7) and cannot be considered a signature. The virion preparations, but not EVs and NVEPs, were also enriched in Lamp1 (Figure 3), a known LE protein that traverses the endosomal system via the CDE route and passes through the EEs/SEs [47]. The presence of Lamp1 suggests a detour of EE flux, as none of the other LE proteins could be detected in the virions. This is consistent with the observations that CMVs congregate in the endosomal compartments traversed by the CIE and CDE cargo, resulting in their retention in the AC [30]. 

The virion preparations were not enriched in the CDE cargo proteins that take the EE route to LEs/lysosomes (such as CD63 [51], LDLR, and EGFR [47]), traffic between the EE and TGN (such as CD63, [51]), and circulate between the EEs, LEs, and TGN via the retrograde recycling route (such as TGOLN2 [TGN38/42], GLUT4, M6PRs, WLS [Wntless], Furin, or Vamp4), or circulate at the interface of EE and LEs (such as NPC1) (Figure 3). The absence of these cargo proteins is consistent with the absence of retrieval machinery (see Section 3.7) and suggests that the retrieval domains in EE are not used for envelopment. The virion preparations also lacked G protein-coupled receptors (GPCRs) (Figure 3), which are retrieved by SNX27 into tubular endosomes and returned to the PM in cooperation with the retromer and the WASH complex (Figure 4). This, together with the absence of SNX27 in the virions (see Section 3.7), suggests that the SNX27-dependent recycling domain does not contribute to envelopment. Thus, CIE and CDE cargo transport in the EEs of CMV-infected cells is arrested after the sorting of degradative and retrograde cargo, and envelopment likely occurs at the membranes traversed by the CIE cargo proteins.

### 3.3. Rab GTPases

Rab proteins are small GTPases that are master regulators of membrane flux in eukaryotic cells [98]. Similar to other small GTPases, they are activated by guanine nucleotide exchange factors (GEFs) to bind GTP (GTP-bound form) and inactivated by GTPase-activating proteins (GAPs) that facilitate the conversion of GTP to GDP (GDP-bound form). Activated Rab proteins are prenylated and inserted into the membrane, and inactivated Rab proteins are extracted from the membranes by GDP dissociation inhibitor proteins (GDI) and stored in the cytosol. The cycles of activation, followed by insertion into membranes, and inactivation, followed by extraction, determine membrane identity. More than 60 Rab proteins, together with seven phosphoinositides, create the navigation tags that control the recruitment of the effector proteins to the membranes, regulating biogenesis, transport, tethering, and the fusion of the membrane organelles. Membrane flux can be viewed as a sequential wave of Rab recruitment and de-recruitment that directs membrane flux in different directions within the cell. Therefore, Rab signatures in CMV virions should shed light on the history of membranes used for envelopment and their biochemical properties in the context of the known functions of Rab proteins in the membrane system.

Rab proteins are highly recruited to the AC area of CMV-infected cells [26], indicating a highly polymorphic membrane composition within the AC. This area is huge and may harbor thousands of membrane-like units in fibroblasts [26], as EM imaging showed that most membrane units are 50–200 nm in diameter [14,29,34,35,37,99]. Although several Rab proteins were expected in the virion preparations, the number of identified Rab proteins was surprisingly high, and at least 16 Rab proteins were detected in the virion preparations treated with proteinase K (Figure 5) Most of these Rab proteins were associated with the tubular domains of the REs, ERC, TGN, and ERGIC. Their presence in the virions would indicate that the virions are enveloped by a complex multidomain membrane structure. However, most of these Rab proteins were also enriched in *l*EVs, *s*EVs, and even NVEPs (Figure 5, proteome 7). Although their enrichment in vesicular and non-vesicular extracellular components does not preclude their incorporation into the virions, most of these Rab proteins cannot be considered a reliable signature within the virions. Among the 16 Rabs detected in the proteinase K-treated virion preparations, Rabs 12, 18, 32, and 23 were either not detected or were present in low amounts in *l*EVs, *s*EVs, and NVEPs (Figure 5, proteome 7). Three of them (Rabs 12, 18, and 23) were consistently reported in other proteomic studies of HCMV virions (Figure 5, proteomes 2–5). Thus, these Rabs have arisen as the likely signature that may reveal the identity of the envelopment membrane.

#### 3.3.1. Rab12, Rab18, Rab23, and Rab32 

Rab12 localizes to the EEs and ERC and controls the transport of specific cargo, such as TfR, from the ERC to lysosomes [125] and stimulates autophagy [131], autophagosome trafficking [132], and retrograde transport from the ERC to TGN [133]. It has been suggested that Rab12 facilitates the direct or indirect fusion of RE/ERC-derived compartments with lysosomes [125] and links autophagosomes to motor proteins to facilitate autophagosome trafficking [132]. 

Rab18 has been localized in the ER and endosomes, and its activity is associated with several processes including autophagy, secretion, and lipid droplet biogenesis (reviewed in [127]). However, its contribution to these processes has not been well characterized, and existing evidence shows that it plays an important role in tethering to the ER and autophagy. Together with Rab10, it regulates ER tubulation, with Rab10 required for ER tubule expansion and Rab18 contributing to the tethering and fusion [134]. The best-established interaction partner of Rab18 is its GEF Rab3GAP [135], which also interacts with ER and the ERGIC protein ERGIC-53 [136], allowing its activation and binding to autophagosome membranes [137]. In addition to the ER, Rab18 is recruited to secretory granules [138], synaptic vesicles [139], and lysosomes [140] in neurons and neuroendocrine cells.

Rab23 has been found in tubular endosomes [67], colocalized with Rab5, and internalized Tf in later stages of TfR trafficking in the early recycling compartment, but not in early stages of the EE pathway, LEs, and Golgi [105]. It also contributes to autophagosome development [141].

Rab32 is localized in the EE /REs, mitochondria, ER, and lysosomes [108,109]. Its activity is associated with SNX6/retromer trafficking, trafficking from the EE/REs to melanosomes, the recycling of VAMP7 from melanosomes to early/recycling endosomes, and the regulation of lysosome-related organelles [109]. It is also localized to the ER and supports autophagic membrane formation and autophagy under basal, nutrient-rich conditions [142].

Overall, all four Rab proteins that pass through the NVEP, *l*EV, and *s*EV filters are associated with membrane system tubulation, ERC function, and autophagy. Their presence in the virions suggests that envelopment may occur on the tubular membranes derived from the CIE recycling pathway, consistent with the identification of the CIE cargo proteins in the virions (Figure 3). It is likely that these membranes belong to the downstream segments of the CIE recycling pathway, as the virions did not contain appreciable amounts of Rabs 22a, 35, and 36 (Figure 5), three GTPases reported to act at a proximal stage of maturation of the CIE recycling pathway (Figure 4) [143]. Rabs 12, 23, and 32 are involved in the exit pathways from the recycling endosomes distinct from the Rab8a/b pathway [67,105,109,125]. However, their role in the endosomal recycling circuit is insufficiently studied, and it is not possible to determine the functional sequence between them or to assign the effector functions associated with their recruitment. Interestingly, all Rab proteins that pass the NVEP filter are associated with autophagosomes, suggesting that the functions acquired by their recruitment may be important for the maturation of autophagosomes, including the development of phagophores. Thus, it is possible that these functions are redundantly activated by the dysregulated recruitment of a “package” of Rab proteins in the downstream tubular compartments of the modified (rearranged) tubular compartments of the inner AC. These membranes can be expanded and used for envelopment.

#### 3.3.2. Rab Interactors and Effectors

Nine Rab proteins (Rabs 1a, 1b, 2a, 5c, 6a, 7a, 10, 11b, and 14) were consistently detected in all the CMV virion preparations (Figure 5, proteomes 1–5). However, these proteins did not pass the *l*EV, *s*EV, and NVEP filters (Figure 5, proteome 7), which does not allow conclusions about their incorporation into the virions. Given that these Rabs act as the master regulators of membrane flux and are highly recruited to the membranes of the AC [26], we also analyzed the presence of their interactors and effectors in the virion preparations.

Rabs 1a, 1b, and 2a are key organizers of the intermediate compartment (IC) [102], a vesiculo-tubular cluster in the secretory pathway between the ER and cis-Golgi that forms several pericentrosomal and peripheral subcompartments located in close proximity to the highly curved tubular-vesicular membrane domains of the ER, known as ER Exit Sites (ERES) [102]. The pericentrosomal IC is often positioned as a linker compartment of the Golgi ribbon with spatiotemporal dynamics that overlap with the ERC [144,145]. This topology and tubular phenotype make the IC a suitable structure for CMV envelopment. The virion preparations are also rich in several components of COPI complexes (Appendix A), which are recruited to the IC by both Rab1 and Rab2 [102] but are also abundant in *l*EVs and NVEPs (Appendix A, proteome 7). Since almost all tethering proteins of IC and other effector proteins of Rab1 and Rab2 [102] are absent in the virions (Appendix A), there is no clue to suggest that IC-derived membranes are used for envelopment.

Rab6a acts at the TGN, where it provides docking of endosome-derived vesicles [146]. It recruits the Golgin tethering factors and nucleates the tethering complexes required for the fusion of the arriving endosome-derived vesicles. Rab6 also regulates intra-Golgi trafficking and mediates retrograde transport from the cis-Golgi to ERGIC or ER via tubular carriers, known as the COPI-independent retrograde pathway [147]. The targeting and fusion of these retrograde carriers are mediated by active Rab18 at the ER membrane through interaction with ER-localized tethering factors [127,135,148]. Rab6a is also involved in the regulation of autophagy [149]. The trapping of Rab6 by enveloping the virions may suggest that envelopment also occurs at the Golgi-tract-derived membranes that form the inner (TGN) and outer (Golgi medial and cis- and trans-Golgi stacks) AC area of the CMV-infected cell [26,150]. However, none of the Golgin tethering factors and components of the COG and GARP tethering complexes (Appendix A) and Rab6a interactors (Appendix A) were found in the virions, suggesting that Rab6-positive TGN-derived membranes do not contribute to envelopment. The exception is the presence of BICD2 (Appendix A), a known Rab6a effector at the Golgi–ERGIC–ER interface.

Rab7a, a key component of the LE pathway, controls transport to LEs and lysosomes and lysosomal biogenesis, positioning, and functions [151]. It is also localized to the ER and modulates ER morphology by controlling ER homeostasis and ER stress [152]. Rab7 is also found on autophagosomes [141]. On endosomes and lysosomes, Rab7a interacts with several effectors, but none of these proteins were found in the virions (Appendix A), suggesting that the envelopment does not occur at the membranes of endosomal or lysosomal origin. 

Rab11b, a member of the Rab11 family, was highly enriched in the virion preparations. It recruits mainly to the membranes of the ERC and contributes to the control of endosomal recycling and the cell surface proteome [121]. It can also be localized in the EEs, TGN, and post-Golgi vesicles. Rab11b, like Rab11a, contributes to the control of the recycling of the CDE and CIE cargo proteins [43] but may act in different ways [153] and have different localization [154]. Rab11b can be recruited to peripheral lysosomes and regulate lysosome exocytosis [123]. However, none of the major regulatory components of lysosome exocytosis (Rab3a, Sec15, and GRAB) [123] were found in the virions.

Rab10 has a variety of functions and subcellular localizations and is involved in various activities in the ERC, Golgi/ TGN and endosomes [155], ER tubulation [156], autophagosome biogenesis, and autophagic flux [157]. Thus, the capture of Rab10 to the enveloping virions may be associated with any tubular compartment, as Rab10 contributes to the initiation of tubulation. The multiple functions of Rab10 have been linked to the recruitment of numerous effectors (Appendix A), but none of these proteins have been found in the virions.

Rab5c, similar to Rab5a and 5b isoforms, is an EE Rab essential for the cell survival and maintenance of EEs and LEs [158]. Rab5c can act semi-independently of the other Rab5 isoforms [159], can be recruited to the Arf6/integrin recycling pathway [160], and drives CD93 and active β1-integrin recycling [161]. Consistent with the capture of CIE cargo and several Rab proteins acting in the endosomal recycling pathway, Rab5c can also be captured by the enveloping virions in this pathway.

In addition to the known effectors and interactors, an important sign for Rab protein recruitment may be the identification of specific GEF and GAP proteins, which should remain as a signature. However, none of the GEF and GAP proteins reviewed by Müler and Goody [130] were found in the virions (Appendix A).

#### 3.3.3. Why Were There So Many Rabs in the Virion Preparations?

The detection of nearly half of the Rab protein repertoire in the virion preparations may be related to their membrane-bound forms within heterogeneous EVs that copurify with the virions. Because of the physiology of the intracellular Rab cycling, Rab proteins could also be collected and detected in nonmembrane-bound forms either in EVs or NVEPs as adjacent cytosolic Rabs associated with chaperone proteins. The switch between the membrane and cytosol is an important mechanism for the regulation of Rabs [130,162]. After inactivation (conversion to a GDP-bound form), many Rabs are extracted from the membrane by GDI proteins and remain bound to the GDI near the membrane to await the next round of activation. Therefore, many Rab proteins detected in the virion preparations may be those associated with GDIs rather than membranes. Indeed, GDI2 was highly enriched in the virion preparations, as well as in *l*EVs, *s*EVs, and NVEPs (Figure 5).

#### 3.3.4. What Did We Learn from Rab Analysis?

Analysis of the Rab proteins in the virions could not answer the question of the origin and identity of the envelopment organelle. Most of the identified Rab proteins passing the *l*EV, *s*EV, and NVEP filters (Rabs 12, 23, and 32) are associated with tubular REs harboring the CIE recycling cargo, consistent with the identification of CIE recycling cargo proteins in the virions. These Rabs suggest that the envelopment organelle develops from membranes derived from REs, consistent with the abundance of RE domains within the inner AC [20,26,163]. However, the identification of other Rab proteins that do not pass the *l*EV, *s*EV, and NVEP filters should not be neglected. Most of these Rabs are also associated with REs (Rabs 5c, 10, 11b, and 14) but some of them are also associated with tubulation at the ER and ERGIC (Rabs 6a, 7a, 10, and 18) and might suggest that these tubular domains are used for the establishment of the envelopment organelle. The bulk of the ER and ERGIC membranes are dislocated outside the inner AC [19,20,22,26], but there is insufficient information on the presence of ER- and ERGIC-derived membranes in the inner AC. The identified Rab proteins do not rule out a contribution from the TGN tubular domains and weakly support the possibility that the envelopment organelle could derive from the vacuolar EE domains abundant in the inner AC [19,23,26,28,163,164] or from LE domains, which are extruded from the inner AC [19,22,26,163].

All of the Rabs found in the virions are associated with autophagosome biogenesis and maturation, suggesting that autophagosomal membranes can be used for envelopment. However, as discussed later (Section 3.10), none of the key autophagic factors were identified in the virions.

### 3.4. Arf GTPases

The Arf superfamily includes 6 Arfs, 22 Arls, and 2 Sars that control a wide range of cellular functions [114]. All members of the Arf subfamily were found in the virion preparations (Figure 5, proteomes 1–6). However, Arfs 1, 3, 4, and 5 were substantially present in NVEPs, *l*EVs, and *s*EVs, whereas Arf6 was highly enriched in *l*EVs and *s*EVs (Figure 5, proteome 7). Only five members of the Arl subfamily were detected in the viral proteomes (Figure 5, proteomes 1–6) but none of them were detected in convincing amounts after proteinase K treatment (Figure 5, proteome 1). Both members of the Sar subfamily were detected in the HCMV preparations and Sar1a was present in convincing amounts (Figure 5, proteome 1) but did not pass the NVEP and *l*EV filters (Figure 5, proteome 7).

Unfortunately, the incorporation of Arf proteins into the virions is not a suitable signature. All Arf proteins can be activated at different sites within the membrane system (Figure 5, [113,114,115,118]) and their recruitment into the virions can indicate any membrane structure within the system. In addition, Arf proteins are highly entrapped in EVs and NVEPs, which may complicate their detection in the virions. Nonetheless, Arf proteins appear to be of prominent importance in the biogenesis of the AC and possibly in the secondary envelopment of beta-herpesviruses and virion egress. All Arf proteins have been shown to be highly recruited to membranes within the AC of MCMV-infected cells: Arf3 at the outer and all Arfs at the inner AC membranes [26,165]. Class I (Arfs 1 and 3) and class II (Arfs 4 and 5) Arfs are also overexpressed in the early phase of MCMV infection and remain elevated in the later stages during virion assembly [165]. The suppression of Arf1, Arf3, Arf4, or Arf6 functions by siRNA prevented the establishment of the pre-AC in MCMV-infected cells [165], suggesting their significant contribution to membrane organelle remodeling. However, the suppression of Arf1 and Arf6 also inhibited the establishment of infection [165], suggesting that these GTPases are also involved in the earliest stages of infection.

Several studies on HCMV- [30,164,166] and MCMV- [25,26,27,97,165] infected cells have shown that the CIE and CDE cargo are retained inside the AC and their recycling is inhibited. These observations support the identification of the CDE and CIE cargo proteins within the virions (Section 3.2 and Figure 3). The inhibition of the cargo exit from the endosomes of CMV-infected cells has been associated with the alteration of Arf6 function and may provide the basis for the envelopment organelle formation. Immunofluorescence studies of HCMV [30] and MCMV [25,26,27,165] showed excessive recruitment of Arf6 to the membrane units within the inner AC, beginning early in infection. This region of the AC is the site of the secondary envelopment of CMV and is composed mainly of EE -, ERC-, and TGN-derived elements. The HCMV study [30] demonstrated that the CIE cargo is stacked in enlarged Arf6 endosomes that retain Arf6 in the GDP-bound form.

Arf6 acts mainly on tubular endosomes that mediate endosomal recycling [167] and contributes to the regulation of autophagy [167,168,169]. These tubular endosomes, known as the tubular endosomal network (TEN) [67,68,170], recycle a subset of CIE cargo proteins to the PM (i.e., CD44, CD98, and CD147) [43,44]. The entire process requires the sequential and orchestrated activation of Rab5, Rab10, Rab22a, Rab35, and Rab11, culminating in the activation of Arf6 (ARF6:GTP) and its hydrolysis to Arf6:GDP before the recycling carriers are released toward the PM [43,44,66]. Another subset of cargo proteins (i.e., MHCI, CD55, and CD59) moves downstream in the EE tract, accumulates in EEA1-positive vacuolar endosomes, and is sorted into the tubular extension to be packaged into transport carriers [44,66,171]. Some of these carriers are returned to the PM via the TEN, whereas others are transported to the juxtanuclear cluster of tubular endosomes that form the ERC [44,46]. Transport to the ERC requires Rab11 activity and the CIE cargo proteins are sorted into Rab8/Arf6/Rab35 intermediates [172]. At the Rab8/Arf6/Rab35 endosomes, Rab35 can recruit multiple Rabs (i.e., Rabs 8, 13, 36) that move on in different directions [111]. Rab8 carriers with recruited Arf6 turn off Rab35 [111,173] and migrate to the PM to ensure the recycling of the CIE cargo from the ERC [111]. It is unclear to what extent the branching can occur at this stage. It is known that Rab36 is required for the delivery of carriers with cargo to the TGN [111], but also that Rab12 [125], Rab23 [67,105], and Rab32 [109] may be recruited and can contribute to different trafficking routes. Thus, multiple recycling paths may be associated with the Arf6-dependent recycling route, and there is still a long way to go to fully understand how the recycling cargo is sorted at this point [70]. In the end, Arf6:GTP hydrolysis is required for recycling and if Arf6 is not inactivated, the endosomal carriers do not reach the point of further progression, the cargo is not recycled, and the endosomal compartment enlarges and retains the endocytic cargo [43,44], as occurs in CMV-infected cells.

### 3.5. Rho Family GTPases

Some virion preparations were also enriched in RhoA, Rac1, and Cdc42 (Figure 5), the three canonical members of the Rho family GTPases, which consists of 22 genes encoding at least 25 proteins [126]. The Rho family is involved in all cellular processes that depend on the organization of the cytoskeleton and plays a role in the regulation of vesicle transport and endocytosis. However, they were also highly enriched in *l*EVs, *s*EVs, and NVEPs (Figure 5, proteome 7) and, therefore, cannot be identified as host cell signatures.

### 3.6. Adaptor Protein Complexes for Cargo Sorting

The formation of coated vesicles for transport between the membrane compartments is orchestrated by heterotetrameric adaptor protein complexes (AP) [174,175]. They recruit cargo proteins by binding to their cytoplasmic tails and facilitate the formation of clathrin-coated vesicles on membranes. They are activated by cargo and clathrin recruitment and by membrane-associated proteins such as Arf1 and FCH domain-only (FCHo) proteins [176]. Adaptor protein 1 (AP1) complexes can be activated at the TGN, EE, LE, and RE/ERC and can mediate the transport of cargo between the TGN and LE in both directions, from the TGN to the PM, and from the ERC/RE to the PM (recycling). AP2 complexes are associated with endocytic activities at the PM, AP3 likely mediates transport from the TGN to LEs and from LEs to lysosomes, AP4 is involved in transport from the TGN to EEs, and AP5 complexes regulate transport from LEs [174].

Almost all components of AP1 and AP2 complexes were found in the virion preparations (Figure 6, proteomes 1–6) and AP2 components were retained after the proteinase K treatment (Figure 6A, proteome 1). However, the components of AP1 and AP2 complexes were highly enriched in *l*EVs, *s*EVs, and NVEPs (Figure 6A, proteome 7). Similarly, clathrin heavy chains and dynamin 2 were highly enriched in most of the virion preparations [10,11,12] but also in *l*EVs, *s*EVs, and NVEPs [16] (data not shown). Nevertheless, the AP complexes, clathrin, and dynamin may play an important role in CMV assembly as they are enriched at the membranes of the inner AC of MCMV- [24,26] and HCMV- [29,163,177] infected cells.

In addition to classical adaptors, clathrin assembly can be facilitated by so-called monomeric adaptors; GGA complexes (Golgi-localized, γ-ear-containing, ADP-ribosylation factor-binding proteins) associate with the TGN and Hrs associate with endosomes [178]. Arrestins (ARRB1-2), epsins (EPN1-3), DAB2, ARH (autosomal recessive hypercholesterolemia protein), and several other proteins can also link cargo to clathrin and can therefore be classified as alternative adaptors [179]. None of these proteins were found in the proteinase K-treated virion preparations.

Overall, the analysis of the adaptor complexes provides no evidence that they can be used as a sorting mechanism for the concentration of viral glycoproteins at the envelopment membranes. Furthermore, this analysis suggests that the envelopment membranes are not active in cargo sorting based on clathrin recruitment. 

### 3.7. Sorting Nexin Code in Virions

Sorting nexins (SNXs) are proteins containing a phosphoinositide-binding phox homology (PX) domain that are transiently recruited to specialized and restricted areas of membranes by a variety of mechanisms [56,61,180,181]. The SNX family includes 34 members divided into six subfamilies based on their domain organization [180]. Among the largest subfamilies are the SNXs that contain only a PX (PX subfamily) or a PX and BAR domain (PX-BAR subfamily). Their main role is to form complexes that control cargo sorting, membrane deformation, and interaction with the cytoskeleton and motor proteins.

SNX functions are associated with the PM, where they contribute to endocytosis, or with endosomes, where they regulate endosomal tubulation and cargo retrieval for transport to the PM (endosomal recycling), TGN (retrograde transport), or to lysosomes (degradation) [180]. The endosomal tubulation function is associated with membrane deformation properties mediated by several mechanisms, but mainly with SNXs containing a curvature-sensing BAR domain (SNXs 1, 2, 4, 5, 6, 7, 8, 30, and 32) [56,180]. The cargo retrieval function is related to their ability to recognize different sorting signals at the cytoplasmic domain of a cargo protein [56,181,182], leading to the cargo sorting to a specific part of the endosomal membrane [56,76,79]. This process is assisted by the development of protein complexes required for cargo concentration in an endosomal microdomain and is termed cargo retrieval [76,183]. The major retrieval complexes are the retromer and retriever and in certain cases also CCC (CCDC22, CCDC93, and COMMD) or WASH (Wiskott–Aldrich syndrome protein and SCAR homolog) complexes and branched actin [56,76]. Retrieval by these complexes enables massive transport toward the TGN, which occurs in the vacuolar domain of EEs and requires support from effector functions recruited by the GTPase Rab7 [76,181]. Cargo retrieval in the recycling domain of EEs involves the formation of tubules in the EE microdomains through the Rab- and Arf-orchestrated cascade-like recruitment of effector proteins that drive the overall process of tubule expansion and fission [56]. SNXs contribute to these processes by facilitating the tubulation and sequestration of cargo in these tubules. Even more, by developing microdomains on the endosomes, SNXs can drive the maturation of the endosomal system, as recently shown by the contribution of SNX1/4 in the FERARI (factors for endosome recycling and Rab interactions) complex-dependent maturation of the Rab11 recycling pathway [184]. The contributions of SNXs in EE maturation are shown in Figure 4.

The tubulation and retrieval functionalities of SNXs may be of particular interest for the development and maintenance of the membranes used for the envelopment of beta-herpesviruses. The enveloping membranes should be enriched in virus-encoded glycoproteins that form the viral envelope, and their concentration at the enveloping membrane should be based on the cargo concentration principles, which is a major function of SNXs. Therefore, the SNX code should remain at the membrane, be captured by the enveloping virions, and be identified in the virion preparations. As shown in Figure 6B, proteomic analysis of the virions detected several SNXs. SNX2, SNX3, and SNX9 were consistently enriched and were also identified in the proteinase K-treated virion preparations (Figure 6B, proteome 1). However, SNX2 and SNX9 were identified in EVs and NVEPs (Figure 6B, proteome 7).

The identification of SNX2 and SNX3 may suggest that tubulation and retrieval functions at the vacuolar EE domain are used for envelopment. These SNXs, together with SNX1, SNX5/6, SNX12, and the retromer and retriever, mediate cargo sorting and transport to the TGN (Figure 4). Two components of the retromer (VPS35 and VPS29) were present in the virion preparations but were also highly enriched in EVs and NVEPs (Appendix A). None of the components of the CCC and WASH complexes (Appendix A) and retrograde cargo proteins (Figure 3) were identified in the virions. SNX8, which can regulate the EE-to-TGN route through a pathway other than that of the SNX1-retromer complex (Figure 4) [79,90,94], was also not detected in the virions (Figure 6). Thus, it appears that the retrieval functions of the vacuolar EE domain are not exploited for the establishment of the envelopment organelles.

In the recycling domains of EEs, there are several pathways for SNX-dependent cargo sorting (Figure 4). The best-known are retromer-based sorting by SNX27 and retriever-based sorting by SNX17 [56,76]. In addition to the retromer, these pathways also recruit the CCC and WASH complexes [76]. Neither SNX27 and SNX17 nor components of the retriever, retromer, CCC, and WASH complexes were convincingly found in the virion preparations (Figure 6 and Appendix A), suggesting that these retrieval pathways at the EE recycling domain are not utilized for the concentration of the virion envelope proteins.

In addition to retromer/retriever-associated sorting for recycling, there are also indications that SNXs can mediate cargo sorting without association with these large complexes. SNX1/4, SNX3, SNX9, and SNX16 are involved in the development of microdomains that form recycling tubules and transport cargo to the PM (Figure 4). SNX4 and possibly SNX1 form a large complex known as FERARI, which is required for cargo sorting into the Rab11-dependent pathway and the ERC [184,185], whereas SNX16 supports the tubulation of EEs [186]. The absence of SNX1, SNX4, and SNX16 (Figure 6) and the major components of the FERARI (Appendix A) in the virions suggest that the EE domains shaped by these proteins do not contribute to envelopment. Although SNX9 was highly present in all of the virion preparations, its contribution to the envelopment organelle cannot be considered since it acts mainly in the PM-related processes [79] and was highly present in EVs and NVEPs (Figure 6). 

SNX3 remained the most important signature that can provide information about the envelopment organelle. Membranes containing SNX3 are highly enriched in the inner AC area of MCMV-infected cells [33]. This SNX has only the PX domain and binds to membranes via an interaction with phosphatidylinositol-3-phosphate [PI(3)P] [187], which is essential for its recruitment, as acute depletion of PI(3)P leads to complete dissociation of SNX3 from membranes [26,33]. Although PI(3)P membranes are dispensable for the biogenesis of AC, PI(3)P itself is essential for the release of infectious MCMV virions [33].

SNX3 can act in several subdomains of EEs (Figure 4) and to a lesser extent in REs [188]. Its function in the vacuolar EEs is mainly associated with the retromer whereby SNX3 contributes by recognizing a structural motif in the cytoplasmic domain of cargo proteins. SNX3 does not contain a BAR domain and therefore does not bend the membrane by itself. However, recent studies have shown that the SNX3-retromer can form an arch-like structure on membranes and cause membrane bending [189]. SNX3 can also be recruited to tubular endosomes independently of Vps35, a component of the core retromer complex, and sort cargo itself [70] or interact with cargo (i.e., TfR) together with Vps35 to form a non-classical retromer complex and thereby sort cargo into REs [190]. Therefore, SNX3 in the virions suggests envelopment at the EE-derived recycling tubules. These tubules may belong to the Rab8/Arf6 recycling pathway (Figure 4), as a recent report indicates an essential role of SNX3 in the formation of Arf6-associated recycling tubules and the recycling of some CIE cargo proteins [70]. 

Some SNX3-mediated CIE recycling tubules do not converge to classical EEA1-labeled EEs, suggesting that SNX3 may also play a role in earlier stages of EE recycling (Figure 4). SNX3 may be suitable for sorting viral proteins into these tubules because the Øx[L/M/V] sorting motif recognized by the SNX3-retromer is relatively simple and present in most viral proteins embedded in the virions. However, this sorting pathway does not appear to be utilized during CMV envelopment because recognition of this motif requires the coincident interaction of cargo proteins, PI (3)P, SNX3, and a component of the retromer core complex VPS26 [189] that was not detected in the virions (Appendix A). 

Overall, analysis of SNXs in the virions suggests that beta-herpes virions can assemble at the SNX3-positive membranes derived from the Rab8/Arf6 recycling pathway and that an SNX3-based cargo sorting mechanism can be used to concentrate viral envelope glycoproteins at the envelopment membrane. 

### 3.8. SNARE Proteins and Tethering Complexes

Almost every membranous compartment has the ability to generate specific labels for vesicle docking and fusion and to generate specifically marked vesicles to direct transport to another compartment [191]. The molecular “zip codes” for these events are primarily regulated by Rab proteins and phosphoinositides that recruit tethering factors (reviewed in [192]), which mediate the initial recognition of transport vesicles by the target membrane and establish target specificity. More than 20 tethering factors are categorized as homodimeric coiled-coil proteins or multisubunit tethering complexes, which capture vesicles over longer or shorter distances, respectively [148,193]. Tethering is followed by SNARE (soluble N-ethylmaleimide-sensitive factor attachment protein receptor)-mediated fusion [194,195,196]. The SNARE protein family, which in humans has 36 members (Figure 7A) [196], provides the targeting signal and, together with the Rabs and phosphoinositides, determines the destination of the transport vesicles. Unlike Rabs, which are transiently recruited to spatially restricted areas, SNAREs are anchored in the membrane and should be returned to the donor compartment via the transport pathways after fusion [196]. Therefore, they may overlap with other SNAREs in membrane compartments as “passengers” during transport to a suitable destination.

More than 12 multisubunit tethering complexes, built of at least 57 cellular proteins and at least 10 homodimeric tethering factors, recruit to the membrane organelles of the endosomal and secretory pathways (Appendix A). None of these subunits was found in the virion preparations, suggesting that virion envelopment does not occur at the membranes that accommodate incoming vesicles. On the contrary, of the 39 proteins in the SNARE group (Figure 7A), VAMP2, VAMP3, STX12, VTI1A, and SEC22B were highly enriched in the virions, even after treatment with proteinase K (Figure 7, proteome 1). VAMP3 and STX12 were consistently found in all HCMV (Figure 7A, proteomes 1–5) and HSV-1 (Figure 7A, proteome 6) proteomes. Most of the SNARE family members (except for STX3, STX4, and SNAP23) were not found at significant levels in EVs and NVEPs (Figure 7A, proteome 7) and therefore might represent a reliable host cell signature.

The fusion of the transport carriers to the acceptor compartment requires the assembly of R-SNARE proteins present at the membrane of a transport carrier (known as v-SNARE) with a trimolecular complex consisting of Qa, Qb, and Qc-SNARE proteins (also known as t-SNAREs) (Figure 7C). In some cases, Qb and Qc can be replaced by the Qbc proteins. The pairing of v- and t-SNARE is tightly regulated, and multiple combinations of R- and Q-SNARE proteins, known as cognate SNAREs, function together in a given transport pathway. The SNARE complexes also mediate the homotypic fusion of membranes, whereby both membranes should contain assembled tetrameric complexes consisting of R- and Q(a,b,c)-SNARE proteins (Figure 7C) [195]. Prior to fusion, these complexes should be disassembled at both membranes to allow SNARE protein assembly between the membranes.

VAMP2 and VAMP3 are R-SNAREs that can be delivered to the envelopment membranes by membrane carriers or by homotypic fusion and serve as the label transport carriers generated from these membranes. VAMP2 is located in the secretory vesicles, PM, EE, RE /ERC, and lysosomes and is involved in the neurosecretion and recycling of synaptic vesicles [194]. It mediates homotypic fusions at EEs through complex formation with STX12/13 (Qa) and SNAP25 (Qbc) [196]. VAMP3 localizes to EEs and REs and regulates transport to other EEs and the PM, Golgi, and autophagosomes [194]. At the autophagosomes, it mediates fusion between autophagosomes and MVBs to generate amphisomes [197]. It is required for the recycling of TfR, GLUT4, and integrins; acts at Arf6 recycling endosomes; and mediates the Arf6-dependent fusion of EEs/REs with the PM [196]. Endosomal sorting of VAMP3 is regulated by its coupling with Vti1a (Vps-ten-interacting-1a) [198], a Qb-SNARE that locates at endosomes and the Golgi and mediates homotypic EE fusion, recycling from EEs/REs, retrograde LE-to-TGN transport, and EE-to-TGN fusion [196,199].

STX12 (also known as STX13 and STX14) is considered to be an EE syntaxin along with STX7, STX8, and STX11 [200]. STX12/13 is Qa SNARE, which traffics the PM-EE-RE/ERC route and is mainly retained at the membrane domains within the EE and RE/ERC [196] and regulates endosomal recycling [201,202]. It forms complexes with VAMP2 and VAMP3 at the branched and partially interconnected membranes of tubular EEs and REs [201]. STX12/13 thus belongs to the t-SNAREs located at the membranes of EEs and REs that accommodate vesicles containing VAMP2 and VAMP3 v-SNAREs. It also forms a complex with Vti1a (Qb) and STX6 (Qc) to interact with VAMP4 (R), thereby mediating the homotypic fusion of EEs [196]. Complex formation with Vti1a is involved in the homotypic fusion of phagophores to form autophagosomes [203]. 

SEC22B is a nonessential R-SNARE, which is mainly found in the ER, ERGIC, cis-Golgi, and autophagosomes [196,204]. It may play a role in transport between the ER/ERGIC and phagosomes. It is a marker of secretory autophagosomes and is essential for the fusion of autophagosomes with the PM [205].

Therefore, the presence of VAMP2, VAMP3, STX12/13, and Vti1a in the virions is further evidence for the envelopment of virions at membranes derived from REs. These SNAREs may either be trapped as passengers or selectively enriched at the envelopment membranes to provide functions required for the envelopment or post-envelopment events. Since STX12/13 and VAM2/3 are cognate SNAREs that act at multiple sites [196], they are likely delivered to the envelopment compartment by vesicular transport or by the homotypic fusion of endosomes. SNX12/13 (Qa) may be complexed with Vti1a (Qb) but the Qc or Qbc SNARE partners were not consistently found in the virion preparations, suggesting that the envelopment membranes could carry incomplete tSNARE complexes. The lack of functional Q-SNARE complexes could be involved in nonproductive side reactions and contribute to futile cycling. For example, STX1 when alone forms dimers and tetramers [195]. The disassembly of the SNARE complexes is mediated by the NSF (N-ethylmaleimide-sensitive fusion factor) and α- SNAP (soluble NSF attachment proteins; α- SNAP is also known as NAPA) complexes, which may promote futile cycling [195]. Both proteins were present in the virion preparations, but they were also enriched in *l*SVs and *s*SVs (Figure 7B). In normal SNARE physiology, the uncoordinated SNARE complexes are disassembled by the chaperone function of the tethering complexes and SM proteins (Sec1-Munc18 proteins), which prevent futile cycling and the development of unphysiological SNARE complexes. None of the components of the tethering complexes (Appendix A) and members of the four major families of SM proteins (Figure 7B) were enriched in the virion preparations, suggesting that an important function of SNARE chaperoning may be absent at the envelopment membranes.

### 3.9. Phosphoinositides and Lipid Code in Virions

Phosphoinositides (PI) are also an important tool for cells to code membranes (reviewed in [206]). By phosphorylating phosphatidylinositol at various sites, the cell controls the membrane association of regulatory and effector proteins, ensuring proper membrane domain composition. Phosphatidylinositol-4,5-phosphate [PI(4,5)P2] controls PM-associated events, PI(3)P drives the maturation of EEs, and phosphatidylinositol-4-phosphate [PI(4)P] controls maturation within the Golgi stacks and TGN. The PI transformation reactions are controlled by the regulated and sequential recruitment of PI kinases and phosphatases to membranes. Therefore, PIs, PI kinases, and PI phosphatases may also represent the code of the envelopment membrane found in the virions.

The distribution of PIs in the AC was studied in MCMV-infected cells using immunofluorescent PI-binding probes [33]. The inner AC membranes were enriched in PI(3)P [33], whereas the outer AC membranes were enriched in PI(4)P (Figure 8A), although PI(4)P membranes could also be found in the inner AC area. This distribution is consistent with the distribution of other membrane organelle markers [26].

Of the 19 PI kinases and 28 PI phosphatases [206,207,208], only 2 were consistently identified in the HCMV and HSV-1 virion preparations: PIK3C2A and OCRL (Figure 8B and Appendix A). PIK3C2A is a member of the class II phosphatidylinositol 3-kinases (PI3K) that phosphorylates PI and PI(4)P, leading to the synthesis of PI(3)P and PI(3,4)P2, respectively [207]. It drives Rab11 activation at the EEs by generating a localized burst of PI(3)P, which is required for the release of the vesicles that transport the membranes toward the ERC [209]. OCRL is the PI(3)P phosphatase that plays a role in the cargo transport from EEs to the TGN and cargo recycling to the PM [210]. Thus, PIK3C2A and ORCL act at EEs and REs, providing further evidence that these membranes are a site of envelopment. None of the class I PI3Ks, which act mainly at the PM and peripheral EEs [206], were found in the virions (Appendix A). PIK3C3 (known as Vps34) was also never identified in the virion preparations (Appendix A). This kinase is a master regulator of the vacuolar Rab5-dependent part of the EE pathway [206], supporting the conclusion that the vacuolar domain of EEs does not contribute to beta-herpesvirus envelopment. The PI4Ks were occasionally found in the HCMV virions (Appendix A), suggesting that the PI(4)P-dependent membrane domains do not contribute significantly to CMV envelopment.

In addition to building the molecular machinery required for membrane adaptation, the envelopment organelle should adapt the lipid composition that favors the envelopment process. The tubulation of membranes is accompanied by the accumulation of phosphatidylserine (PS), a cylindrical phospholipid unfavorable for developing curvature. The membranes of the inner AC in the MCMV-infected cells were highly tubular and enriched in PS, as evidenced by the excessive recruitment of Evectin-2 (also known as PLEKHB2) [26], a PS-binding protein that binds to REs [211]. Glycolipid analyses of the HCMV-infected cells and HCMV virions [42] showed that the virion envelope is rich in phosphatidylethanolamine (PE) and poor in PS, phosphatidic acid (PA), and PIs. This composition resembles the lipidome of the synaptic vesicles, suggesting that the envelopment occurs at specialized cellular membranes with an exocytic capacity [42]. The reduction in PIs suggests that the envelopment occurs at membranes distant from intense PI conversion reactions. The lipidomic data are supported by the proteomic analysis of the virion preparations, which showed an enrichment of the PE-binding protein PEBP1 [212] and an absence of the PS-binding protein PLEKHB2 (Figure 8C). However, PEBP1 was also highly enriched in *l*EVs and *s*EVs (Figure 8C, proteome 7) and therefore may not accurately reflect the PE enrichment in the virions. PE is an inverted-conical lipid that can promote the development of the negative curvature at the membranes required for the formation of the virion envelope around the tegument matrix and cytoplasmic capsids [42,213]. Therefore, the envelopment membrane is expected to adjust the glycerophospholipid composition by the accumulation of PE at the cytosolic leaflet, which can be achieved at the endosomal membranes mainly by the decarboxylation of PS and translocation between leaflets [214]. However, there is no signature for these activities in the virions, as none of the cellular proteins that contribute to these processes (i.e., PS decarboxylase, scramblases, and flippases) were detected in the virion preparations [10,11,12,13,14]. This is not surprising as these reactions could occur on the non-enveloping side of the membrane organelles and may be completed earlier in the process of membrane adaptation. Overall, the lipid codes in the virions suggest that the envelopment occurs at the later stages of the tubular recycling membranes when most PIs have already been degraded and a large fraction of PS has been converted to PE. 

### 3.10. Signatures That May Relate to Autophagy

Membrane flux in the cell is also used to organize specific processes such as autophagy, ciliogenesis, or cytokinesis. CMV infection reorganizes almost the entire membrane system including these processes. Autophagy is a highly conserved intracellular process in which cytoplasmic aggregates, nanoclusters, nano-organelles, and membranous organelles or their components are engulfed by an isolation membrane that expands and wraps around to form an autophagosome (reviewed in [215,216,217]). The autophagocytic material is enclosed in a double membrane that matures and eventually fuses with lysosomes to degrade the entrapped contents. Some autophagosomes fuse with EE-derived MVBs that deliver material to the PM and release the contents outside the cell [205,218]. 

The EM images of CMV envelopment are reminiscent of autophagosome growth [28,34]—the nucleocapsids are wrapped by double membranes. Therefore, rescheduling autophagosome biogenesis or the use of autophagosomal pathways may be an obvious target of CMV-induced changes to ensure appropriate membranes for the secondary envelopment. Several studies have shown that HCMV infection affects autophagy (reviewed in [28,219]). It initially activates the formation of autophagosomes but later blocks their fusion with lysosomes. These changes are among the early events of biogenesis prior to the AC and have been observed very early during infection [28]. The suppression of autophagic flux reduced the virion production, and autophagic proteins could be detected in the purified extracellular virions, suggesting that membranes driven by the autophagic machinery contribute to the envelopment process [28]. Therefore, one should expect to find evidence of autophagic machinery in the virion preparations.

Autophagosome formation can occur at several sites, including the ER, Golgi, endosomes, and ERC [215,216,217]. The prevailing view is that autophagosomes are initiated at omegasomes, an ER-associated membrane compartment. After initiation (also called nucleation), the double membrane begins to expand to form the phagophore, which expands to envelop the autophagic material and closes (known as sealing) to form the autophagosome, a double-membrane vesicle/vacuole. The formed autophagosomes migrate to the cell center and fuse with the lysosomes (known as maturation), resulting in the degradation of the autophagic material. The entire autophagosome cycle involves the sequential recruitment of the complexes organized by the ATG proteins (autophagy-related genes) and ends with the lipidation of the LC3 protein, a known marker for autophagosomes. Thus, if phagophore or autophagosomal membranes were used for beta-herpesvirus envelopment, at least some of the signature proteins of the initiation and elongation processes would be found in the virions. The key autophagic factors contributing to all steps of autophagosome biogenesis are listed in Appendix A. None of these proteins were found in appreciable amounts in the virion preparations (Appendix A). Therefore, it is unlikely that phagophores derived from the ER and autophagosomal membranes are used for CMV envelopment.

The extensive elongation of membranes during phagophore expansion requires a supply of lipids and membrane material, which can occur by several mechanisms including vesicular trafficking from the Golgi, PM, and REs. All these processes are supported by the recruitment of Rab GTPases and effector proteins, including lipid-modifying and curvature-promoting proteins. Almost all of the Rab proteins identified in the virions are associated with EE/RE trafficking and may play a role at various stages of the autophagosomal pathway. Phagophore initiation and growth require the production of PI(3)P, whereas phagophore envelopment around the autophagic cargo requires PE for the development of the negative curvature [216]. All of these lipid modifications have been identified in the lipidomes of autophagic membranes [220] and purified HCMV virions [42], as described in Section 3.9.

Autophagic flux also involves the formation of amphisomes [205,218], a hybrid organelle that migrates and either fuses with lysosomes or with the PM to provide an unconventional secretion. However, the available information is insufficient for distinguishing this secretory pathway from the exosomal pathway. The amphisome step requires additional machinery, such as Rab GTPases, tethering complexes, SNAREs, and motor proteins, which are recruited concomitantly with elongation and possibly entrapped within the virions during envelopment. Amphisome development requires Rab8a [221] and Rab11 and involves RE tubules that contribute to autophagosome formation [141]. The maturation of autophagosomes into amphisomes involves the STX17 [222], STX6, VAMP3 [223], or YKT6 [224] SNARE proteins and two major tethering complexes HOPS and CORVET [218]. With the exception of VAMP3, none of these components were found in the virions (Figure 6 and Figure 7 and Appendix A), suggesting that amphisomes are not the site of envelopment. 

Overall, the absence of key autophagic factors in the virions suggests that beta-herpesvirus envelopment does not occur at the membrane organelles formed by the autophagosomal pathway. However, most of the identified signatures suggest that the envelopment organelle shares functional properties with phagophores and autophagosomes. 

## 4. Discussion and Conclusions

Analysis of cargo proteins, small GTPases, adaptor proteins, sorting nexins, tethering complexes, SNARE proteins, and phosphoinositide kinases and phosphatases in the available proteomes of extracellular herpesvirus virion preparations indicates several host cell proteins that may be signatures of the biogenesis history and functionalities of the cellular membrane organelles used for envelopment. These host cell proteins emerged after extensive filtering of the proteins present in *l*EVs, *s*EVs, and NVEPs, extracellular particles that have recently been resolved by a combination of high-resolution density-gradient fractionation and direct immunoaffinity purification [16] or asymmetric flow field-flow fractionation [17,18]. The presence of the proteins in *l*EVs and *s*EVs does not preclude their incorporation into the virions. However, higher-resolution studies are needed to resolve the virions from different forms of EVs and to approach the host cell signatures that these extracellular particles may share. Also, the extracellular vesicles and likely virion preparations were laden with relatively large amounts of non-vesicular extracellular particles (NVEPs). 

All of the available published proteomes of the herpes virion preparations [9,10,11,12,13,14,15] contained many cellular proteins associated with NVEPs, as revealed by proteomic analysis of NVEPs isolated from the supernatants of two cell lines [16]. In the study by Turner et al. [10], a substantial portion of these proteins was removed by proteinase K treatment. Nevertheless, many proteins that remained after proteinase K treatment were also found in NVEPs, suggesting that more extensive purification of the virions is needed to reveal the host cell signatures hidden in the virions. NVEPs are heterogeneous matter containing various cytoplasmic nanoparticles that are engulfed by autophagosomes and released from cells after amphisome fusion with the PM [16]. 

Many of the components found in NVEPs have been found in the proteome of the ER and may be related to ER stress. Of the 238 proteins significantly modulated after tunicamycin-induced ER stress in HeLa cells [225], 61 were found in the proteinase K-treated virion preparations and 45 of them were upregulated after ER stress. Moreover, all of the virion preparations contained many ER-resident proteins, both membrane-associated and intraluminal. Thus, the ER stress-induced release of the material may contribute to NVEPs and even membrane-enveloped particles that copurify with the isolated virions. Therefore, it is difficult to exclude the ER as a source of the membrane organelles used for CMV envelopment. Several studies on the HCMV [6,19,20,22,226] and MCMV [26] AC have suggested ER dislocation from the area where the enveloping virions have been observed. However, given the emerging discoveries of multiple ER contact sites with other membrane organelles [227] and the active role of the ER in shaping the membrane organelles, as shown by the contribution of the ER in the scission of the tubular endosomes [228], higher-resolution studies are needed to investigate the role of the ER membranes in the envelopment process. Membrane organelles such as the ER and Golgi can also be engulfed and processed into amphisomes that fuse with lysosomes for degradation [218]. Although there is as yet no direct evidence for the release of membrane elements by amphisomes, it is likely that this pathway is used by many cells. This would be expected especially after ER stress and the Unfolded Protein Response (UPR) of the cell, which occur during infection with beta-herpesviruses [219]. Moreover, infection with beta-herpesviruses is associated with severe contraction and shrinkage of the infected cell, and secretory autophagy of membranes may be an important mechanism for regulating cell volume and membrane content. Therefore, to identify the host cell signatures in the proteomes of the virion preparations, it would be important to screen out ER-associated host cell proteins that might be released from infected cells.

After extensive filtering out of the *l*EV-, *s*EV-, and NVEP-associated proteins, we narrowed down the identification of the host cell signatures in the HCMV virions (Figure 9). The presence of MHC-I proteins, Rab proteins (Rab12, Rab23, and Rab32), sorting nexins (SNX2 and SNX3), SNARE proteins (VAMP2, VAMP3, STX12, and VTI1A), and the PI(3)P kinase PIK3C2A suggests that the membranes used for envelopment are derived from the tubular extensions of EEs that recycle cargo via the Arf6/Rab8-dependent recycling pathway.

The signatures present in the virions can provide us with information not only about the biogenesis of the envelopment compartment but also about the functionalities used during the envelopment of the capsids. The presence of SNX3 could indicate a cargo sorting mechanism required for the concentration of viral glycoproteins in the envelopment organelles. PIK3C2A, a PI3 kinase that generates localized PI(3)P bursts at the tubular EE membranes after the detachment of a major PI(3)P producer (Vps34) at the vacuolar EE membranes [209], may provide an important source of the conical phospholipid required for membrane growth and membrane attachment of the PI (3)P-dependent SNXs required for viral glycoprotein concentration. The absence of tethering proteins suggests that the enveloping part of the organelle does not receive membrane flux, although this says nothing about the uptake capacity of the outer membrane. The presence of cargo proteins suggests that these organelles are transport-active during biogenesis and awaiting envelopment. The presence of small GTPases and SNARE proteins suggests that they may be active in the recycling pathway. Also, the presence of the SNARE proteins VAMP2, VAMP3, STX12, and VTI1A suggests the existence of a SNARE pool at the membrane that may be utilized in the envelopment and post-envelopment events. These SNARE proteins are likely maintained in a dissociated state by the SNARE-dissociating complex organized by the NSF and NAPA proteins and can be released when needed. For example, these complexes may be released at the growing edges during envelope formation and used for the envelopment membrane closure. Indeed, a knock-down of a VAPM3 study [10] demonstrated that VAMP3 is essential for HCMV replication in various cell types. After closure, the enveloped virions can be released by ESCRT-III-mediated scission. The components of ESCRT-III were not found in the virion preparations, but this does not preclude their contribution because the topology of activation does not guarantee that they will be entrapped in virions. Several studies have indicated the important role of ESCRT in HCMV assembly and egress [229,230,231]. A similar principle of STX12/13-dependent membrane closure and ESCRT-III-dependent scission has been described for autophagophore maturation [203]. As described for amphisome biogenesis [197,218,223], VAMP3 can be used in the envelopment organelle for fusion with MVBs, providing the exit pathway for the enveloped virions. 

Interestingly, many of the host cell signatures present in the CMV virions are associated with phagophore development and autophagosome formation. Because the virions did not contain proteins that mark the autophagosomal pathway, it is unlikely that this pathway provides membranes for envelopment. Thus, the envelopment of the virions requires functionalities of endosomes that can be upgraded with functionalities of engulfment and closure to form the enveloped virions, as well as functionalities that would guide the enveloped virions to a proper site for the exit out of the cell through the multivesicular compartments, which have been reported to be the major pathway for the release of infectious beta-herpes virions [14]. The sorting and concentration of viral glycoproteins may occur through endosomes, which have cargo sorting mechanisms such as SNX3. Viral glycoproteins can be recruited to endosomes by SNXs and collected in endosomal tubular extensions, which mainly sort CIE cargo and are retained in these tubules by inhibiting their conversion to recycling carriers. The viral preparations were rich in CIE cargo transported by this route, and both HCMV [30,164,166] and MCMV [27,97] infection inhibits the recycling of CIE and CDE cargo. Furthermore, the study by Zeltzer et al. [30] has shown that this defect is associated with the prolonged association of Arf6:GDP at the endosomal tubules within the AC.

Overall, the analysis of the host cell signatures within the HCMV virions provided a clear hypothesis about the possible membrane organelles used for envelopment. This hypothesis can be tested in both HCMV- and MCMV-infected cells, as it is likely that the envelopment mechanism is evolutionarily conserved in beta-herpesviruses and even in herpesviruses in general. Experimental approaches to test this hypothesis would require the establishment of unique biomarkers for the envelopment sites and optimized use of biochemical and genetic tools. Because the HCMV replication cycle is quite long (5 days), the use of genetic tools is a major challenge. In contrast, the replication cycle of MCMV in cell culture is much shorter (1 day). Therefore, the combined approach of HCMV and MCMV infection conditions can solve many problems in CMV envelopment. In addition, the establishment of unique biomarkers for the envelopment organelle could greatly enhance efforts to better understand not only the biogenesis of the AC but also the plasticity of the endomembrane system at the EE–RE–TGN interface. This area of the cell is extremely complex and dynamic, and CMV may be an important cell biologist to help elucidate it. 

## Figures and Tables

**Figure 1 ijms-23-09994-f001:**
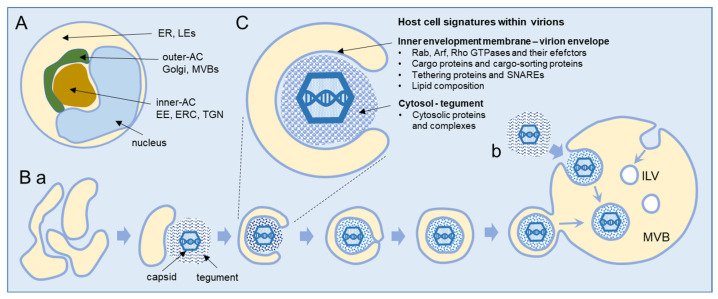
Schematic outline of the beta-herpesvirus secondary envelopment. (**A**) Schematic presentation of the infected cell and the AC. (**B**) Suggested sequence of HCMV cytoplasmic maturation in human foreskin fibroblasts recorded by advanced electron microscopy [2,34]. Secondary envelopment occurs when partially tegumented capsids contact the membranous compartment. This contact deforms the membrane, which wraps around the tegumented capsids and fuse to form enveloped capsids within the double membrane (a), or capsids bud into a large compartment resulting in fission of enveloped capsids into the lumen (b). At a later stage of infection, capsids were observed in large vacuoles, and HCMV virions may incorporate into multivesicular bodies (MVBs). ILVs, intraluminal vesicles. (**C**) Host cell signatures that may be captured into virions during the envelopment process.

**Figure 2 ijms-23-09994-f002:**
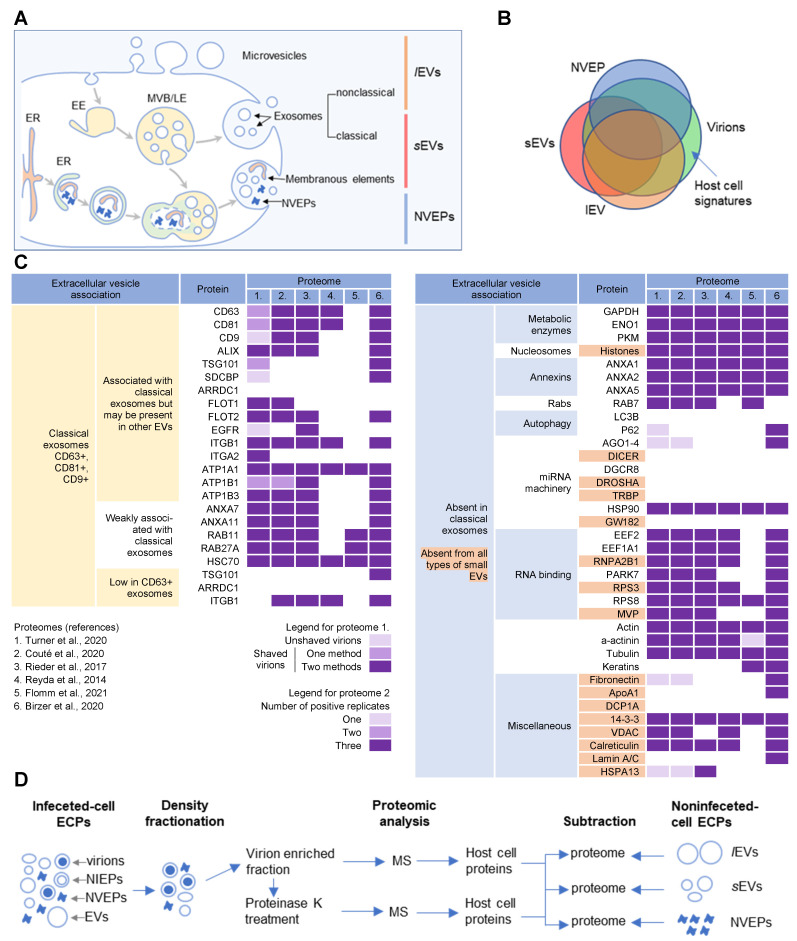
Extracellular nanoparticle content within preparations of beta-herpes virions. (**A**) Schematic representation of extracellular nanoparticle biogenesis via plasma membrane (PM), exosome, and amphisome-dependent pathways. Microvesicles and non-classical exosomes segregate as large (l) extracellular vesicles (*l*EVs), whereas classical exosomes segregate as small (s) extracellular vesicles (*s*EVs). Non-vesicular extracellular particles (NVEPs) and vesicular elements are released by amphisome-dependent and exosome-independent secretion [16,17,18]. (**B**) Schematic representation of the expected proteomic overlap between *l*EVs, *s*EVs, NVEPs, and extracellular virion preparations. (**C**) Reassessed nanoparticle content [16] identified in HCMV and HSV-1 extracellular virion proteomes. Proteins were identified in five proteomes of HCMV virion preparations (proteome 1–5) and one proteome of HSV-1 heavy particles (proteome 6). Proteome 1 [10] presents data for untreated HCMV virion preparations (unshaved) and the same preparation treated with proteinase K (“shaved” virions). Identification of proteins in shaved virions is marked with different color codes depending on whether they were detected by one or two mass spectrometric methods. Proteome 2 [11] displays data in different color codes depending on whether a protein was identified in one, two, or three biological replicates. Proteomes 3–6 (proteomes 3 [12], 4 [13], 5 [14], and 6 [15]) indicate whether a protein is present or not. (**D**) Workflow for identification of host cell signatures from proteomes of virion preparations. All proteomes [10,11,12,13,14,15] generated by the MS analysis of virion-enriched fractions after density-gradient separation contained many proteins that are present in EVs and NVEPs including one subjected to short proteinase K treatment [10]. Proteins identified in significant amounts in the proteomes of *l*EVs, *s*EVs, and NVEPs [16] are subtracted, which are defined as the *l*EV, sEV, and NVEP filters.

**Figure 3 ijms-23-09994-f003:**
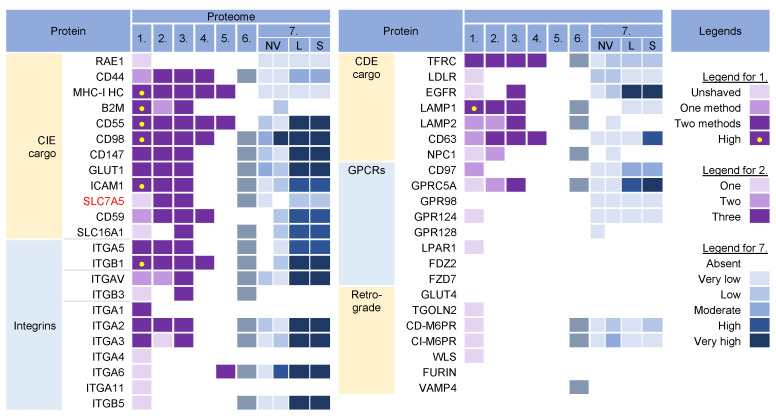
Identification of cargo proteins that travel the endosomal pathways in virion and extracellular nanoparticle preparations. Clathrin-independent endocytosis (CIE) [43,44,45,46] and clathrin-dependent endocytosis (CDE) [47,48,49,50,51,52] cargo proteins, integrins (both CIE and CDE proteins) [53], G protein-coupled receptors (GPCRs) [54], and cargo proteins of the retrograde EE-to-TGN pathway [48,55] were identified in five proteomes of HCMV virion preparations and one proteome of HSV-1 heavy particles as described in Figure 2. Proteome 7 [16] shows the abundance of a cargo protein in high-resolution density-gradient-purified non-vesicular (NV) samples of DKO-1 cells (left box) and Gli36 cells (right box), *l*EV (L), and *s*EV (S) samples of DKO-1 cells. A color code of very low (less than 2 × log_2_ abundance relative to average signal) was set up as a threshold [16]. Low means 2–3×, moderate 3–4×, high 4–5×, and very high ≥5×.

**Figure 4 ijms-23-09994-f004:**
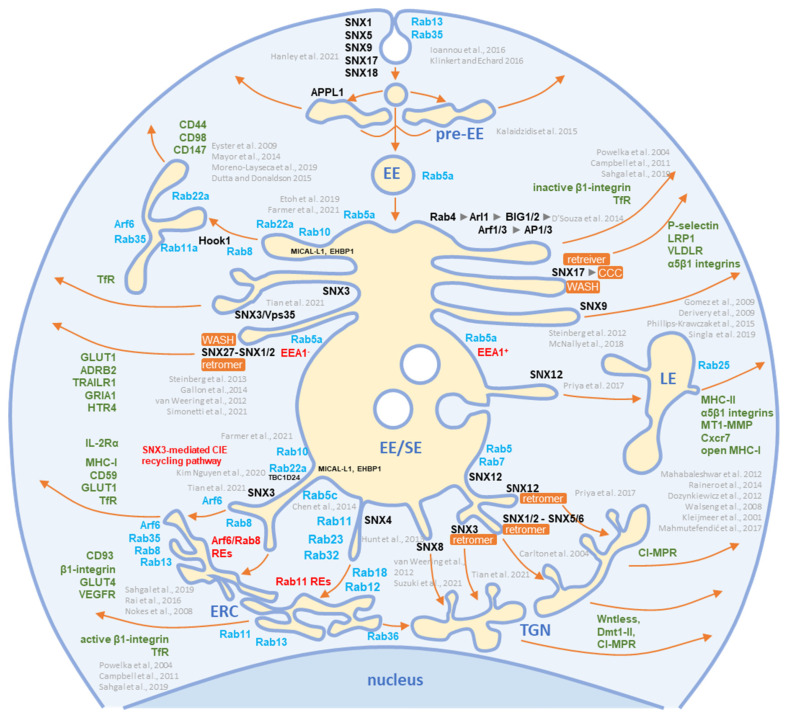
Schematic representation of known exit sites and recycling pathways from the early endosomal system. Cargo internalized by clathrin-dependent endocytosis (CDE) and clathrin-independent endocytosis (CIE) is collected in pre-early endosomes (pre-EEs) and either returned to the plasma membrane (PM) by rapid recycling or delivered to early endosomal (EEs) carriers. EEs undergo a series of fusion and maturation reactions that funnel cargo toward the cell center into enlarged early/sorting endosomes (EE/SE). The maturing EE/SEs sort transported cargo into tubular extensions forming a tubular endosomal network (TEN) that transports cargo directly to the plasma membrane (PM) for recycling, to pericentriolar clusters of tubular endosomes known as the endosomal recycling compartment (ERC) for indirect recycling to the PM, to the trans-Golgi network (TGN) for retrograde transport, or to late endosomes (LEs). EE/SEs also sort cargo by reverse budding into intraluminal vesicles (ILV). Upon completion of the sorting reactions, the remnants of EE /SEs enriched in ILVs become multivesicular endosomes (known as multivesicular bodies, MVBs), which undergo further LE maturation events and either fuse with lysosomes (degradation pathway) or release ILVs as exosomes (exosomal pathway). Cargo sorting into the tubular extension involves the complex and orchestrated recruitment of cellular proteins of the Rab and Arf families, their GEFs and GAPs, members of the sorting nexin (SNX) family, sorting complexes (i.e., retromer, retriever, CCC, and WASH), and other effector proteins required for endosomal tubulation and scission of transport carriers (e.g., dynamins, EHD proteins, AP complexes). The complex cascade of their recruitment is not yet fully elucidated, and the known locations in the EE/SE biogenesis are shown in the schematic. Rab and Arf family proteins are marked in blue, SNXs in black, and cargo molecules in green. Based on references [43,44,46,59,60,61,62,63,64,65,66,67,68,69,70,71,72,73,74,75,76,77,78,79,80,81,82,83,84,85,86,87,88,89,90,91,92,93,94].

**Figure 5 ijms-23-09994-f005:**
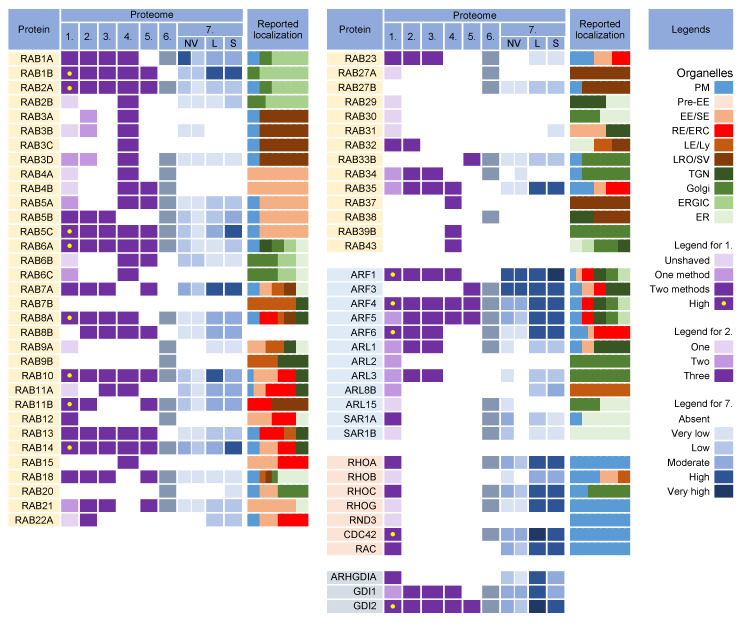
Membrane-trafficking-associated small GTPases in virion and extracellular nanoparticle preparations. Small GTPases were identified in proteomes of virion preparations (HCMV proteomes 1–5 and HSV-1 proteome 6), NVEPs (NV), *l*EVs (L), and *s*EVs (S) (proteome 7). The reported localization in membrane organelles of non-infected cells is shown by a color code to illustrate the preferred localization [49,59,60,69,95,96,97,98,99,100,101,102,103,104,105,106,107,108,109,110,111,112,113,114,115,116,117,118,119,120,121,122,123,124,125,126,127,128,129,130]. Proteomes used for analysis and legends are described in Figure 3.

**Figure 6 ijms-23-09994-f006:**
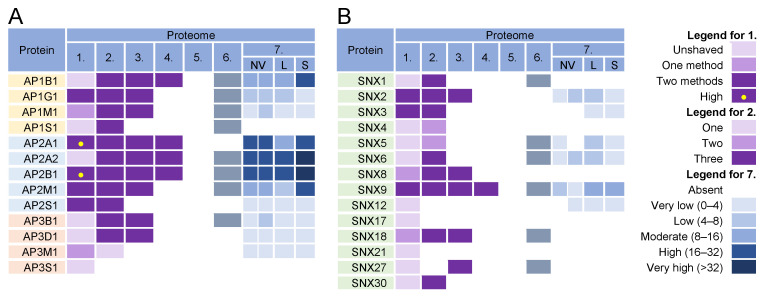
Identification of (**A**) adaptor protein (AP) complexes and (**B**) sorting nexins (SNX) in virions and NVEPs in extracellular vesicle preparations. The proteins are identified in five proteomes of HCMV virion preparations (proteome 1–5), proteomes of HSV-1 heavy particles (proteome 6), and NVEPs (proteome 7), as described in Figure 3.

**Figure 7 ijms-23-09994-f007:**
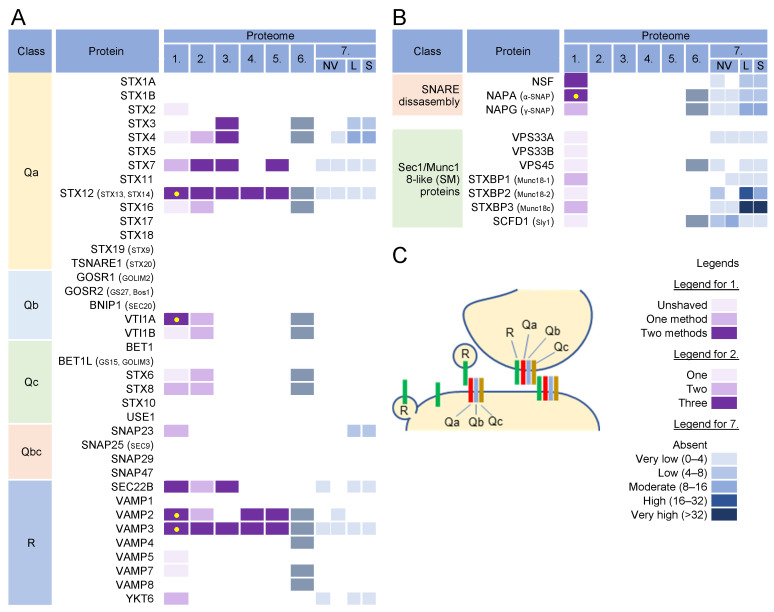
SNARE proteins and components of the SNARE disassembly machinery in virion and extracellular nanoparticle preparations. (**A**) SNARE proteins grouped based on their domain topology [196] and their identification in five proteomes of HCMV virion preparations (proteomes 1–5); proteomes of HSV-1 heavy particles (proteome 6); and NVEP (NV), *l*EV (L), *s*EV (S) nanoparticles (proteome 7), as described in Figure 3. (**B**) Proteins that control and chaperone SNARE assembly [195] and disassembly, and their identification in proteomes of virions and extracellular nanoparticle preparations. (**C**) Principles of SNARE assembly during vesicle fusion with acceptor compartment, homotypic membrane fusion, and development of outgoing carriers [195]. Legend for protein identification in proteomes 1, 2, and 7, as described in Figure 3.

**Figure 8 ijms-23-09994-f008:**
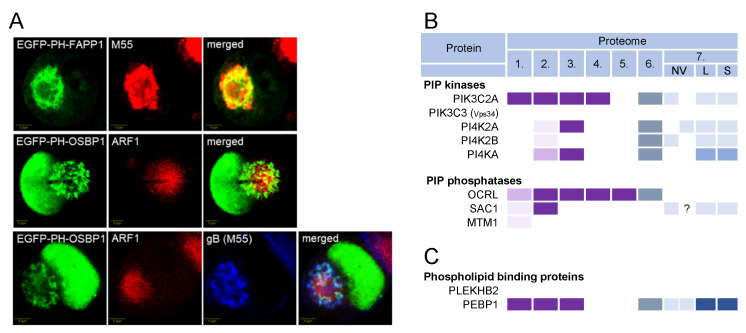
Phosphoinositide codes in CMV-infected cells and virions. (**A**) Images of MCMV-infected cells showing the distribution of the PI(4)P-binding modules (EGFP-PH-FAPP1 and EGFP-PH-OSBP1), Arf1, and viral glycoprotein B (M55). Cells were stained 48 h postinfection as previously described [33]. PIP kinases and phosphatases (**B**) and phospholipid-binding proteins (**C**) identified in five proteomes of HCMV virion preparation (proteomes 1–5), proteome of HSV-1 heavy particles (proteome 6), and extracellular nanoparticle (proteome 7), as described in Figure 3. A complete list of all kinases and phosphatases is shown in Appendix A.

**Figure 9 ijms-23-09994-f009:**
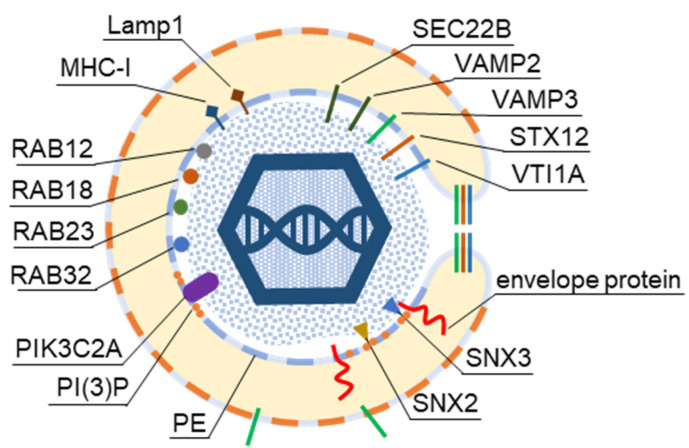
Host cell signatures in HCMV virions that may reveal the envelopment organelle.

## Data Availability

The data presented in this study are available on request from the corresponding author.

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
