# Peer review of "Host Cell Signatures of the Envelopment Site within Beta-Herpes Virions"

_ijms, 2022, doi:10.3390/ijms23179994_

Round 1
Reviewer 1 Report
In the manuscript entitled “Host cell signatures of the envelopment site within beta-herpesvirions” the authors discuss the host cell signatures of the beta-herpesvirion assembly compartments. Below are the suggestions to improve the manuscript.
1. The authors should represent their analysis pipeline in the form of flow diagrams to make it understandable to the readers.
2. The authors should explain the rationale clearly.
Author Response
The manuscript has been revised according to recommendations of the Reviewer #1. The workflow for identification of host cell signatures from proteomes of virion preparations is included as part D of Figure 2. Accordingly, we made appropriate interventions in the main text. Also, we made several interventions in the first part of the manuscript in order to explain the rationale more clearly. All interventions in the text are labeled with the track changes.
Reviewer 2 Report
I have greatly enjoyed, and learned a great deal, reading the review by Lučin et al. The manuscript includes an exhaustive, encyclopedic and well readable work.
In this review, the authors have analyzed published proteomes of human cytomegalovirus (HCMV) virion preparations that identified a large number of host cell proteins. The authors have selected HCMV as a ubiquitous beta herpesvirus and because most knowledge of beta-herpesvirus assembly comes from studies of HCMV. I personally agree with their choice because this virus is a major cause of birth defects, a life-threatening opportunistic infection in immunodeficient individuals, and a potential oncomodulatory agent.
The presented topic matches the journal's scope. The draft is well written and provides substantial graphical details to make reading and understanding easy. Therefore, I do not have any comments to be addressed.
Good luck...
Author Response
Many thanks for supportive comments.